# Exploring the Type and Quality of Peer Feedback in a Graduate-Level Blended Course

Evrim Erbilgin *, Jennifer M. Robinson, Adeeb M. Jarrah, Jason D. Johnson and Serigne M. Gningue

Curriculum and Instruction Division, Emirates College for Advanced Education,
Abu Dhabi P.O. Box 126662, United Arab Emirates
* Correspondence: evrim.erbilgin@ecae.ac.ae

**Abstract:** Most blended graduate courses engage students in peer feedback activities based on sociocultural theories of learning. Despite its growing importance, little is known about the quality and type of feedback that graduate students provide to their peers in online learning settings. This study aimed to explore the quality and type of feedback that graduate students provided to each other during online learning activities that took place in a blended course at a higher education institution in the United Arab Emirates. Volunteer students (n = 24) from four different sections of a graduate course were the participants of this study. The students' feedback to each other in two online discussion forums and in the final assessment paper were the data sources of this study. Qualitative data analysis methods were used to analyze the data. Based on the related literature, we analyzed the data from three perspectives: the type of feedback function (affirmation/negation, justification/explanation, praise, suggestion, question), the level of feedback (task, process, self-regulation), and the quality of each instance of feedback (ineffective, slightly effective, partly effective, effective) to investigate the nature of the peer feedback in detail. The data analysis indicated that graduate students might need support in providing high-quality feedback to their peers. The findings might help instructors improve the existing online or blended course designs.

**Keywords:** peer feedback; online education; blended course; higher education

## 1. Introduction

With the increasing use of technology in education, including the shift from face-to-face to online teaching, instructors are faced with a growing need to implement innovative practices to cope with changes to the delivery of education, including peer feedback practices. Traditionally, students receive feedback only from their course instructors. However, with the increased number of online and blended learning courses, peer feedback is being integrated into the educational context as a practice designed to go beyond the traditional focus on instructors' comments. The need to integrate peer feedback in online interactive learning tasks (e.g., online discussions) is based on the sociocultural theory of learning [1,2], which stresses the role that social interaction plays in psychological and cognitive development. It suggests that human learning is essentially a social process and that our cognitive functions are formed by our interactions with those around us. The rationale behind designing online interactive learning tasks is that this will provide students with a learning environment in which they can co-construct meaning through their interactions with each other [1]. Through this process, the type and quality of feedback that students provide or receive are the keys to ensuring robust learning [3,4]. Students are typically required to read their peers' written responses to assignment prompts and are then asked to provide constructive feedback so that they learn with, and from, each other.

Peer feedback engages students in their own learning process by allowing them to play the roles of both a teacher and student. There is a large body of literature indicating that peer feedback improves students' self-regulation, metacognition, and academic

achievements [1,3–10]. When students provide and receive peer feedback, it is believed that it reinforces their learning and critical thinking skills and enables them to enhance their understanding of the topic by considering different perspectives [11]. However, most of the existing research studies on peer feedback were conducted with K-12 (e.g., [7,12–15]) or undergraduate students (e.g., [5,10,16–18]) in face-to-face learning settings. There are few studies investigating graduate students' learning experiences when they give and receive peer feedback in online or blended learning courses (e.g., [1,3,19–22]). Moreover, little is known about the quality of this feedback in terms of how effective it may be, or about the types of feedback given, such as questions, praise, affirmation/negation, suggestions, or justifications [23]. It is important to understand the nature of peer feedback in online learning environments for graduate students, particularly after the recent increase in the number of online/blended courses due to the pandemic, to develop and design courses that benefit from peer feedback. The current study aims to explore the types and quality of feedback that graduate students provide to each other during online learning activities within a blended learning course at a higher education institution in the United Arab Emirates (UAE).

### 1.1. Peer Feedback

Feedback can be defined as any type of comment on how a student is performing on a task and is given to maximize learning [6,24]. Silvervarg et al. [25] use the term "critical constructive feedback" to indicate the type of quality feedback that helps the learner make progress toward the learning objectives. The related literature reports the positive influence of high-quality peer feedback on students' learning [1,8,26,27]. High-quality peer feedback helps students self-reflect on their understanding and accelerate learning [20,28]. Through giving or receiving peer feedback, students are involved in articulating their emerging comprehension of the subject matter. For example, the feedback provider engages in the cognitive processes of critical evaluation, offering suggestions, and providing explanations [11]. On the other hand, a lack of high-quality and personalized feedback in online learning environments may cause students to feel disengaged in the course activities and may even result in withdrawal from the course [29,30]. Black [31] reported that structure and instructor facilitation are crucial to promoting critical and reflective thinking in the peer feedback process. Otherwise, online communications may tend to be in the form of sharing information without a sign of reflective thinking.

Within the UAE context, there are few studies on peer feedback. Azaza [32], cited in Al-Ghazali [33], investigated the impact of peer feedback on students' writing mechanics at the school level using an action research methodology. The study found that peer feedback helped to improve students' writing skills if complemented with teacher feedback and self-assessment. Hojeij and Baroudi [17] examined undergraduate students' and instructors' perceptions of peer feedback and reported that it improved the students' reviewing and writing skills as perceived by the participants. Similarly, Al-Ghazali [33] explored how peer feedback was practiced among undergraduate students in a traditional classroom setting. The students shared the view that peer feedback is a beneficial process both for the giver and receiver of the feedback in terms of improving learning and increasing confidence. Some students, however, raised concerns about the accuracy of the peer feedback and some others expressed the difficulty of offering criticism to their peers due to cultural norms. A study conducted by Shine [34] found that students prioritized instructor feedback over peer feedback. The research studies in the UAE context mainly focused on students' and teachers' perceptions of peer and instructor feedback. Exploring the type, function, and quality of peer feedback in the Gulf region might contribute to understanding the role of peer feedback in learning settings and contribute to the knowledge base in this field of research.

### 1.2. Higher Education Students Engaged in Peer Feedback

Research has explored higher education students engaged in peer feedback [10,20,21,28]. Various studies have investigated undergraduate students' peer feedback in relation to usefulness and effectiveness. For instance, Berg et al. [35] reported on the peer feedback experience of higher education students who participated in a writing course. The findings focused on the usefulness of the students' peer feedback and, based on the results, three components emerged as critical to the effectiveness of the peer feedback: (1) the assessment task including a formative and summative requirement; (2) groups of students assessing another student's assessment; and (3) students providing oral peer feedback to each other. In other words, when these three conditions are implemented for an assessment, peer feedback was deemed effective by the higher education students. In another study, Ion et al. [36] explored the written peer feedback shared by undergraduate students on a course assignment and students' perceptions of peer feedback. The researchers found that peer feedback focused on solving the task and ways to improve the task and that it promoted motivation for learning and responsibility for self-learning. Students felt that peer feedback facilitated their learning process. It was recommended that a deeper analysis of the nature of peer feedback is warranted to better understand the mechanisms involved in sharing and receiving feedback.

Some researchers focused on peer feedback offered in graduate courses. Ching and Hsu [19] explored the role that gender plays in preferences of discussion modality in a graduate-level online course. The researchers reported that females preferred to give and receive feedback using audio/video while males preferred written communication. Sharp and Rodriguez [21] investigated the effects of the use of two different technology tools on the instructional design of the peer feedback activities of graduate students. The researchers concluded that the type of technology tool that was used in the course for the peer feedback process affected the content of the feedback students provided to each other. Similarly, Yang [22] found that a computer-supported collaborative learning system facilitated an improvement in the writing skills of graduate students and helped them modify their summary writings locally and globally through peer feedback. Within these studies, the focus was not on the function, levels, and quality of the feedback and so these are discussed further in the next subsections.

### 1.3. The Function, Levels, and Quality of Feedback

In the related literature, feedback has been examined and classified in terms of its function, levels and quality [3,4,7,14]. The function of feedback relates to its purpose in terms of activating affective or cognitive domains. Investigating the function of feedback is important as by analyzing feedback comments, the communicative purpose and cognitive aspect of the feedback can be revealed. Based on the related literature [3,6,18,19,37,38], the function of feedback can be grouped under the following categories: (1) affirmation/negation—sharing aspects that the feedback provider agrees or disagrees with the student on; (2) evaluation—providing positive or negative evaluations of aspects of the student's work; (3) suggestion—giving advice on how to improve the student's work; (4) analysis—explaining their own perspectives and arguments drawing from related literature and/or personal experience; (5) question—asking questions to request further clarification and explanation. These categories can be used to examine the function of feedback used in online learning tasks.

When considering the learning process for students, four levels of feedback—task, process, self-regulation, and self—were designed to identify feedback in specific areas [7,14]. Task-level feedback focuses on conveying requirements concerning the correctness or completeness of the accomplishment of the task; process-level feedback targets the strategies or approaches required to perform the task; self-regulation-level feedback assists in student confidence, self-assessment, management, and improvement; and self-level feedback describes positive or negative personal evaluations about the student. It was noted that task-level feedback was considered to be most effective for beginner learners, though

process and self-regulatory levels of feedback were found to be the most appropriate for average and high-achieving learners [39]. Feedback focusing on personal characteristics was not found to be useful to improve learning.

Regarding quality, Hattie and Timperley's [7] research on feedback showed that three questions should guide effective feedback: Where am I going? How am I going? What is my next step? These three questions are important to help students close the gap between their current level of learning and the desired level of learning. According to Hattie and Clarke [14], ineffective feedback focuses on the student themselves who completed the task and does not offer comments on how to improve learning. In contrast, effective feedback provides the students with guidance on how to improve performance and achievement. Characteristics of effective feedback include the following: (1) feedback is offered in a timely manner; (2) comments are clear and specific; (3) feedback is related to task goals, learning objectives, and improvement strategies; and (4) feedback promotes reflection and self-regulation [25,40].

Defining quality feedback as critical constructive comments from others that help the learner make progress toward the learning objectives [25] to ensure robust learning [3,4], this study examined the nature of feedback provided by participants through three different lenses: the function of feedback, the level of feedback, and the quality of feedback. The following research question guided this study:

- What is the function, level, and quality of peer feedback used by graduate students during online learning tasks taking place in a blended course?

## 2. Methodology

This study adopted an exploratory research design [41] to conduct an in-depth analysis of the feedback developed by the participants and shared with their peers. Exploratory research is useful for gaining an understanding of a phenomenon with limited known knowledge. It can be used to discover new aspects of a topic to better understand its nature and characteristics. In our case, we used a qualitative exploratory research approach [42] to further our knowledge of the types and functions of peer feedback used by our students.

### 2.1. Participants

All graduate students enrolled in the mathematics education track of the Post-Graduate Diploma and Master of Education programs at a higher education institution in the United Arab Emirates were invited to participate in this study. All students in both programs were female, having previously completed undergraduate programs, and were currently employed for a minimum of one year in the workforce. Twenty-four (out of 31) graduate students volunteered to take part in this study. The students who did not give consent for participation were still required to complete the learning activities, however, their data were excluded from the data collection and analysis phases. While students self-selected to participate in this study, by requiring all students to complete the learning activities regardless of the research study, aspects of volunteer bias were attempted to be minimized. However, considerations of this for implications of the generalization of the results must be acknowledged.

### 2.2. Data Collection

Prior to data collection, the first author sought and received approval from the Institutional Research Board (IRB) at the Higher Education Institution where they were affiliated in order to commence with data collection. In addition, consent from the participants was sought and gained, with this process administered by instructors other than those teaching each class to avoid undue pressure to participate.

This study collected data from three learning activities that took place in a graduate-level course focusing on assessment in mathematics education. The intention of the course was to upskill graduate students in terms of the use of effective assessment in the classroom and associated mathematical pedagogies. The first four authors designed these learning

activities (ungraded) as they were the instructors of this course, each teaching one section. After the task descriptions for each learning activity were written, each instructor reviewed them for clarity, accuracy, and readability, leading to the refinement of the descriptions. The instructors also discussed and agreed on how to implement the activities for consistency across the sections. The course had eleven 4-h sessions in total and was delivered via a blended approach. For context, the blended approach was defined as a mix of online classes and campus classes [43], where campus classes sometimes utilized a hybrid lesson scenario with the majority of students face-to-face in the campus class, but with some students attending 'online' via the use of cameras and microphones within the classroom setting. Throughout the duration of the course, students attended five sessions that were taught online and six sessions that were taught face-to-face in the on-campus classroom.

The first and second learning activities engaged the students in online discussion forums during the online sessions. In each online discussion, the task incorporated two parts. First, the students were asked to respond to a discussion prompt in writing designed by the instructors, and second, they were required to provide constructive feedback to at least two of their classmates. Figures 1 and 2 show the discussion prompts for learning activities 1 and 2, respectively. The students used class time to respond to the writing prompts and to each other. However, they were able to revisit the discussion forum at a later time during the same week. Students' written feedback to each other in the two online discussions formed part of the data for this study.

> **Assessing Conceptual and Procedural Knowledge**
>
> For this task, choose a mathematics content/topic about which you will write two questions. Write one mathematics question that you can use to assess your students' conceptual knowledge. Explain why you think your question assesses conceptual knowledge. Then, write another question that you can use to assess your students' procedural knowledge. Explain why you think this question assesses procedural knowledge.
>
> Next, read two of your peers' answers to the initial prompt. Please give constructive feedback to your peers. You can write whether you agree or disagree with their question categories, explain your reasoning, recommend revisions, or pose questions to help your peers think deeper about the concepts. Feel free to make connections to the course readings (with correct APA citation)!

**Figure 1.** The discussion prompt for Learning Activity 1.

The third learning activity required students to provide feedback to a peer on one of the course assignments. As part of this assignment, the students developed an assessment tool. The students were asked to give feedback on their peer's assessment tool's clarity, alignment with the learning outcomes, and appropriateness to the grade level. The peer feedback on this course assignment was used in this study.

### 2.3. Data Analysis

This study employed the provisional coding approach within the exploratory coding method [42] to analyze the collected data. In this approach, the researchers start the coding process with a list of codes derived from prior experiences and related literature. During the data analysis process, these codes can be expanded, deleted, or revised. Accordingly, we conducted a thorough literature review prior to the data analysis phase and from this decided to analyze the data from three perspectives to investigate the nature of peer feedback in detail. During the literature review process, we did not limit our search to studies conducted in higher education settings because studies conducted in K-12 settings

offer important insights about feedback. Nevertheless, we made sure to include studies with higher education participants while developing the coding scheme for the three perspectives used in data analysis. The analysis for each perspective allowed us to establish an initial list of codes which were then revised throughout the coding process as appropriate. During the process of the thorough literature review, as described above, we identified three key aspects of feedback: type of feedback, level of feedback, and quality of feedback.

---

**Reflecting on Assessment 2 Questions**

PART A:

1. Upload one open-response question from your assessment tool, with the associated rubric.

2. Write a short description including the following:

    a. Type of question (routine/non-routine)

    b. Focus of question (procedural vs conceptual)

    c. Explain your reasoning/justification for the above.

PART B:

1. Provide feedback to one of your colleagues. Explain if you agree/disagree with their rationale and why.

---

**Figure 2.** The discussion prompt for Learning Activity 2.

First, we analyzed the type of feedback function, drawing on the existing literature [3,6,19,37,38] to find out the purpose of feedback in terms of activating the cognitive domain. The following categories were used to code peer feedback from the feedback function perspective: question, praise, affirmation/negation, suggestion, and justification. Table 1 provides a description of each category as well as an example from the data. For this analysis, each instance of written feedback was unitized into distinct ideas expressed in a phrase, sentence, or paragraph. Then, each unit was coded into one of the feedback function categories. However, when we counted the number of units in each category, we avoided repetition and considered each category's occurrence within one instance of feedback. For example, if a student used two praising comments within one case of feedback, we counted one point of praise for this case of feedback.

Second, we examined the level of feedback using the three categories suggested by Hattie and Clarke [14] and Wisniewski et al. [44]: task, process, and self-regulation. This analysis is important for understanding the cognitive processes that the feedback provider aims to activate. The description of each level with an accompanying example is presented in Table 2. In this analysis, we used a holistic approach and coded each case of feedback into one category only.

Third, using a holistic approach, we determined the quality of each case of feedback using the rubric given in Table 3. This rubric was developed based on the existing literature on feedback [3,6,14,44].

Establishing credibility in qualitative research is crucial for ensuring the trustworthiness of the findings and interpretations. In this study, there are several strategies that were used to enhance credibility. First, the two leading authors coded the data together. NVivo software (release 1.6) was used in the data analysis process. They used the coding scheme explained above during the coding process. In case of any disagreements, the coders used the scheme as a guide for their discussion. All disagreements were resolved, in some cases leading to revisions in the codes. Second, peer debriefing was executed as an additional strategy to establish credibility and increase confidence in their findings. Specifically, the three other authors reviewed and critiqued the data analysis process and the findings. Peer debriefing is a method of seeking input from colleagues to improve the accuracy of the

results. This entails asking a knowledgeable peer to evaluate and analyze the transcripts, methodology, and conclusions. Researchers in qualitative studies employ this approach to examine their work in an objective and impartial manner, thereby reinforcing the validity of their research [45].

**Table 1.** Type of feedback function.

| Type | Description | Example |
|---|---|---|
| Affirmation/Negation | Feedback shares aspects that the feedback provider agrees or disagrees with the student on in a direct or indirect way. | I agree that the question is procedural. (Direct affirmation) |
| Justification/Explanation | Feedback includes explaining their own perspectives and arguments drawing from related literature and/or personal experiences. | ... because usually, students are more familiar with questions that require formulas and steps rather than questions that ask them to explain a statement. |
| Praise | Feedback provides a positive assessment of aspects of the student's work using favorable comments. | Great questions for grade 5. |
| Suggestion | Feedback gives advice on how to improve the student's work. | Maybe provide extra paper so students can cut and paste to compare between triangles. |
| Question | Feedback includes asking questions to request further clarification, or explanation or to promote reflection. | Can you put the units in the multiple choices? |

**Table 2.** Level of feedback.

| Level | Description | Example |
|---|---|---|
| Task | Feedback focuses on the quality of how the task was performed. It may include information on the correctness of the assignment, links between concepts, and the acquisition of more knowledge. | A good question that reflects the student's understanding of the concepts of division, multiplication, and the connection between them. |
| Process | Feedback is about the processes used to complete the task. It may involve a reassessment of the used approaches, offering ways to detect errors, and using different strategies and processes. | I think both examples assess procedural understanding, you could find a non-routine problem to assess conceptual understanding. |
| Self-regulation | Feedback aims at increasing the student's autonomy in their learning. It may contain comments about self-assessment, reflection, and self-management. | [The current data did not include any feedback that fell into the self-regulation category.] |

**Table 3.** Quality of feedback.

| Quality | Description | Example |
|---|---|---|
| Ineffective | Feedback focuses on the student rather than on the task or processes used. Feedback does not provide information about how to improve learning. Feedback includes general comments (praise or criticism) without details. | I agree with you for both questions and reasons. |
| Slightly Effective | Feedback includes information about the task performance or processes used but includes limited details that make it difficult for the student to utilize the feedback to improve their learning. | I agree with you that they can compare and that leads them to do critical thinking. |
| Partly Effective | Feedback involves information about the task performance or processes used. There are justifications, suggestions, or questions that can be used to improve learning; however, these ideas do not promote the use of different processes or self-regulation. | Good example. I agree with you it is routine because it is one step equation. Also, students must know the concept of solving one step equation. But, better if you replace (if not explain her mistake and correct it) by (explain your answer) because your question gave the students the idea that the answer is not correct. |
| Effective | Feedback addresses task goals and the quality of the student's performance with detailed comments. There are justifications, suggestions, or questions that would lead to enhanced learning, the use of different processes, or self-regulation. | Good example: Procedural understanding is when students carry on steps and algorithms and here they follow steps to count units and add numbers to find perimeter and multiply them to find area, so they rarely make deep connections during instruction. I think both examples assess procedural understanding, you could find a non-routine problem to assess conceptual understanding. |

## 3. Results

In this section, we present the findings obtained from the data analysis conducted using three perspectives. These perspectives included the type of feedback function, the level of feedback, and the quality of feedback. There were 30 pieces of feedback in Learning Activity 1, 20 in Learning Activity 2, and 47 in Learning Activity 3, making 97 in total.

### 3.1. Type of Feedback Function

The type of feedback function was determined using five categories. Figure 3 presents the number of feedback responses in each category across the three learning activities. Figures 4–6 show a similar distribution for each learning activity separately.

The data analysis revealed that the affirming/negating, justifying/explaining, and praising types of communication dominated the participants' feedback functions. Out of the total of 97 feedback responses, 81 of them (84%) included affirmation/negation, 63 (65%) included justification/explanation, and 30 (31%) included praise. Typically, the students used justifications/explanations after they praised their peer's work or after they expressed a direct or indirect agreement/disagreement (affirmation/negation) with the peer's work. The following pieces of feedback illustrate this finding:

-   Good example [Praise]: Procedural understanding is when students carry on steps and algorithms and here they follow steps to add numbers using the number line, so they rarely make deep connections during instruction [Justification/Explanation]. (Learning Activity 1).
-   I agree with you it is routine [Affirmation] because it is one step equation. Also, students must know the concept of solving one step equation [Justification/Explanation]. (Learning Activity 2).

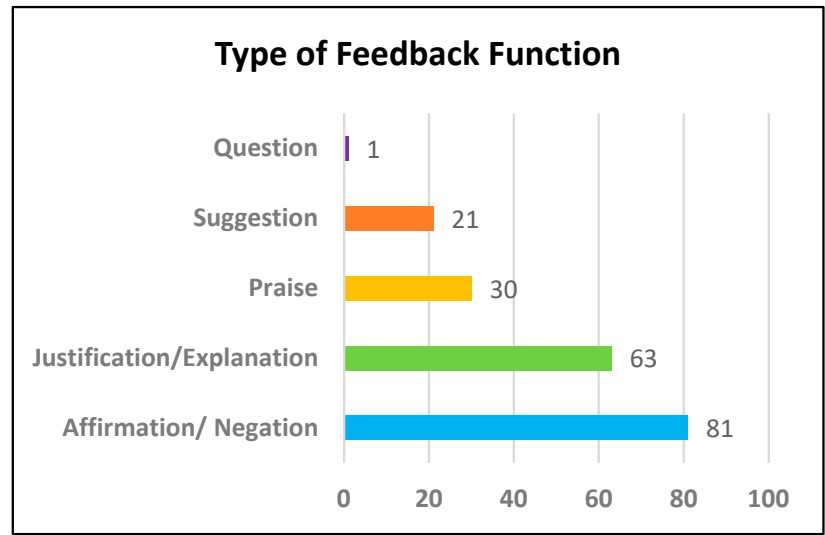

**Figure 3.** Type of feedback function across three learning activities.

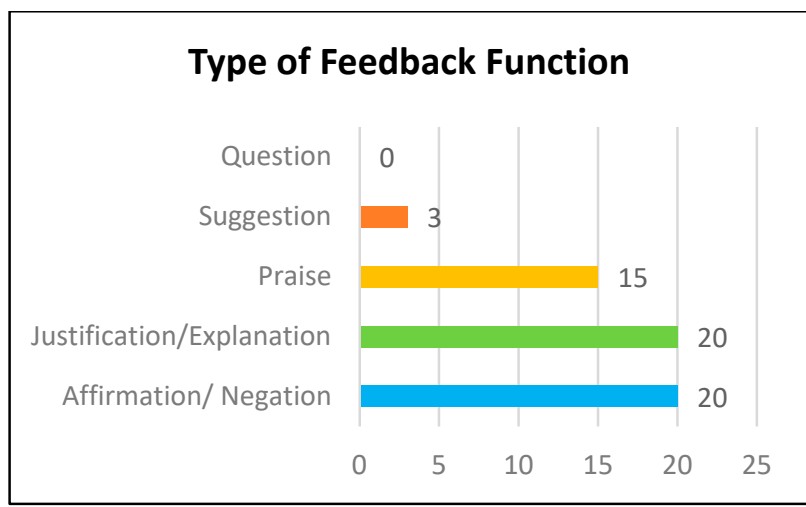

**Figure 4.** Type of feedback function in Learning Activity 1.

The depth of justifications/explanations differed, ranging from an overall comment to elaborating on the peer's work to feed forward. This aspect of the feedback was taken into consideration while analyzing the data from a quality perspective. Some affirming/negating or praising communications were written without justification/explanation:

-   A good example for grade 3 students, and the rubric is clear [Praise]. (Learning Activity 2).
-   This question is not appropriate for grade 3. [Negation]. (Learning Activity 3).
-   The suggesting (22%) and questioning (1%) types of communication occurred the least, compared to the other three categories. There was only one piece of feedback that included a question:

- Yes, it is clear and understandable [Affirmation]. Can you put the units in the multiple choices? [Question] (Learning Activity 3).

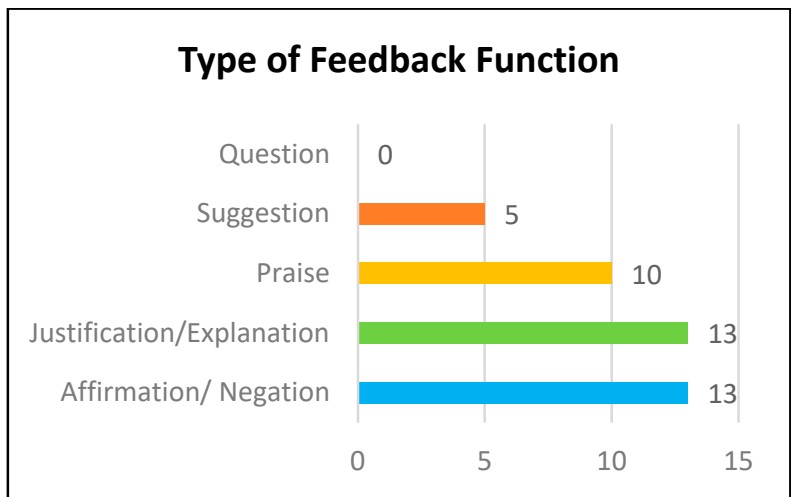

**Figure 5.** Type of feedback function in Learning Activity 2.

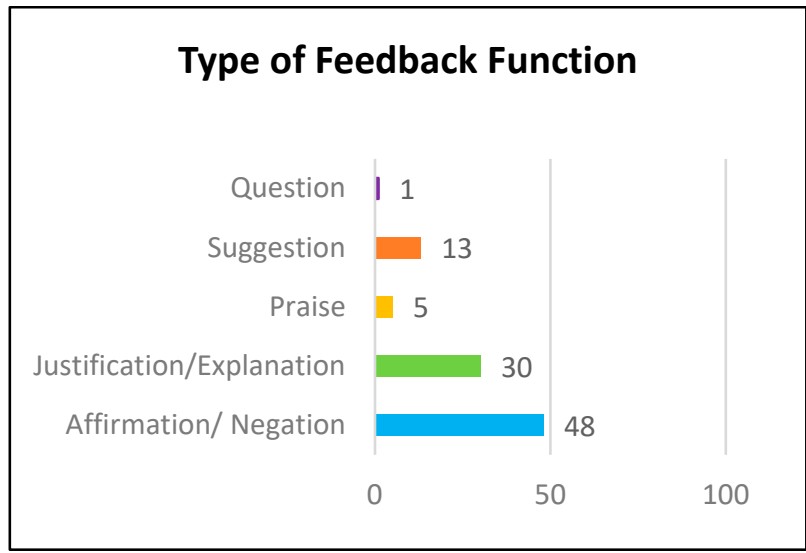

**Figure 6.** Type of feedback function in Learning Activity 3.

This question is a closed type of question, offering a suggestion in the question format. The participating students offered suggestions across the three learning activities to their peers to help them improve their work:

- Perfect example related to conceptual and procedural [Praise]. Even better: Explain the reason why its procedural [Suggestion]. (Learning Activity 1).
- . . . it is a conceptual question [Affirmation] because students can use more than one concept to create this answer. it is a good question! [Praise] provide the rubric for the question please! [Suggestion]. (Learning Activity 2).
- For the question to be more clear change it to the following "Fatima wants to select an excellent location to sell tickets for a graduation ceremony." [Suggestion]. (Learning Activity 3).

The majority of the suggestions focused on the task. The students aimed at improving the quality of the work that their peers completed from an accuracy or completion point of view. In the first two examples given above, the students asked for some additional work (e.g., writing an explanation for why the question is procedural) from their peers as it was

a requirement of the discussion task. In the last example, the student made a suggestion to enhance the clarity of the question that her peer developed.

*3.2. Level of Feedback*

The data analysis process, with respect to the level of feedback, showed that the participants focused on the task while writing their feedback. As Figure 7 presents, 94 feedback responses (97%) concentrated on the quality of how the task was performed. In contrast, three feedback responses (3%) considered the processes used to complete the task. Two of the process-focused feedback responses were written in Learning Activity 1 while the other process-focused feedback response was provided in Learning Activity 2.

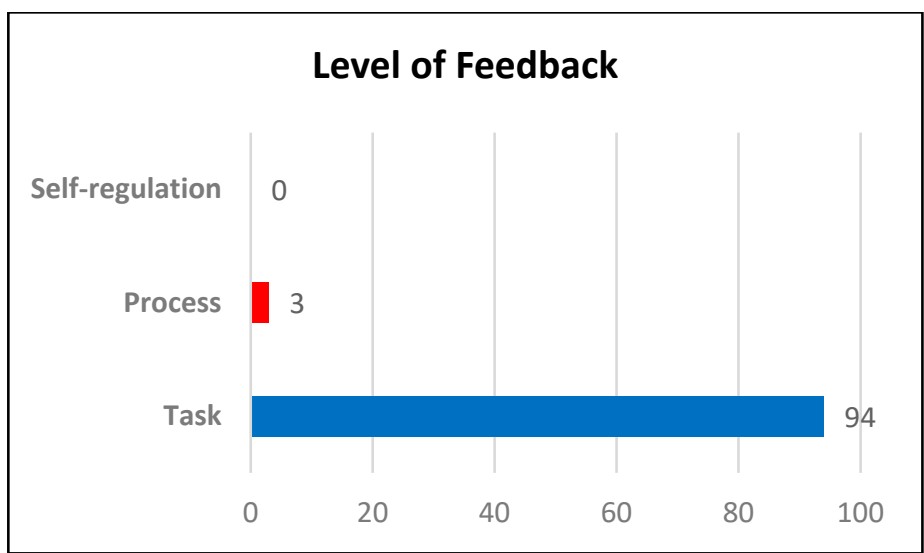

**Figure 7.** Level of feedback across three learning activities.

The feedback focusing on the task involved students' elaborations on the discussion topic, perspectives on the accuracy of the peer's work, and requests for more information or revisions. Examples include the following:

- I agree with you because the example for conceptual involves understanding the relationship between the graph and the radius and procedural question includes operation and action sequence for solving problems. (Learning Activity 1).
- Modify the wording of question 3 by clarifying the phrase 'cubes of numbers of each cube consisting of 6 numbers' so it does not cause confusion to the student. (Learning Activity 3).

The feedback focusing on the process included suggestions on how to complete aspects of the learning task with a different strategy. An example, shared below, from Learning Activity 2 shows a response where students were asked to share an open-response question along with a rubric to assess student performance and an analysis of their open-response question. In the following piece of peer feedback, the student is suggesting her peer use a different type of rubric to better assess student understanding. The rubric shared by the peer was a generic type of self-assessment rubric.

- Good question. But the rubric is a self-assessment type, I think it is better to change it to three levels of understanding. (Learning Activity 2).

*3.3. Quality of Feedback*

The data analysis with respect to the quality of feedback revealed that out of 97 pieces of feedback, one (1%) was categorized as effective. In addition, 25 (26%) were partly effective, 50 (51%) were slightly effective, and 21 (22%) were ineffective. Figure 8 shows the

distribution of feedback across the three learning activities while Figures 9–11 show the distribution for each learning activity separately.

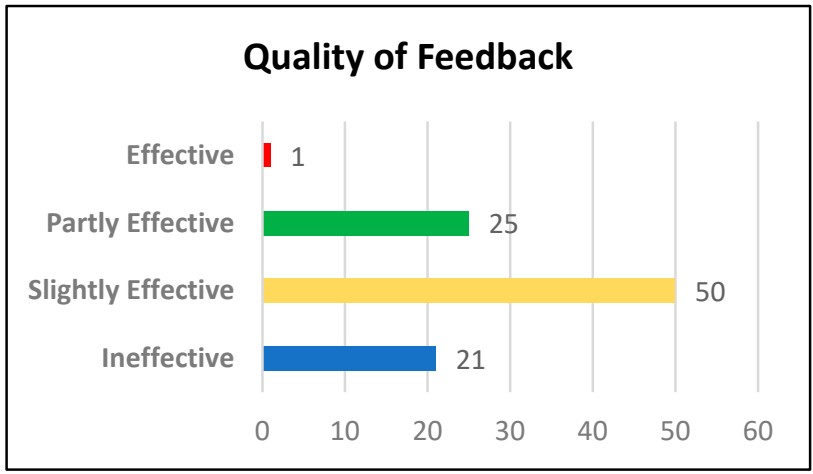

**Figure 8.** Quality of feedback across three learning activities.

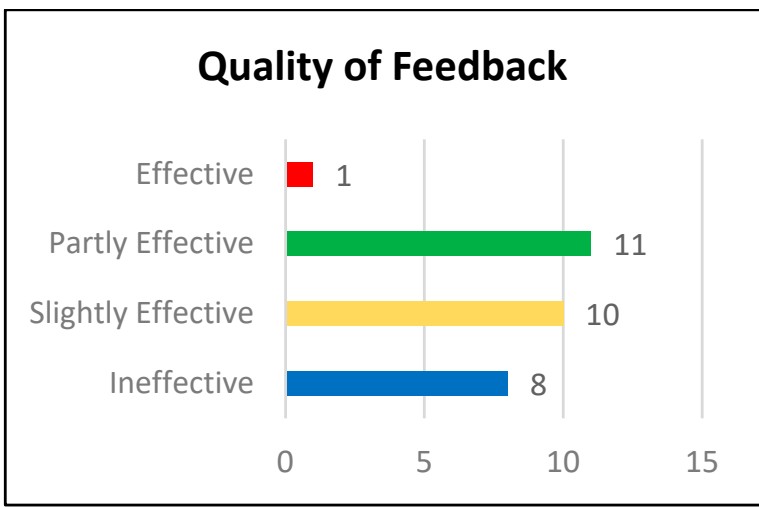

**Figure 9.** Quality of feedback in Learning Activity 1.

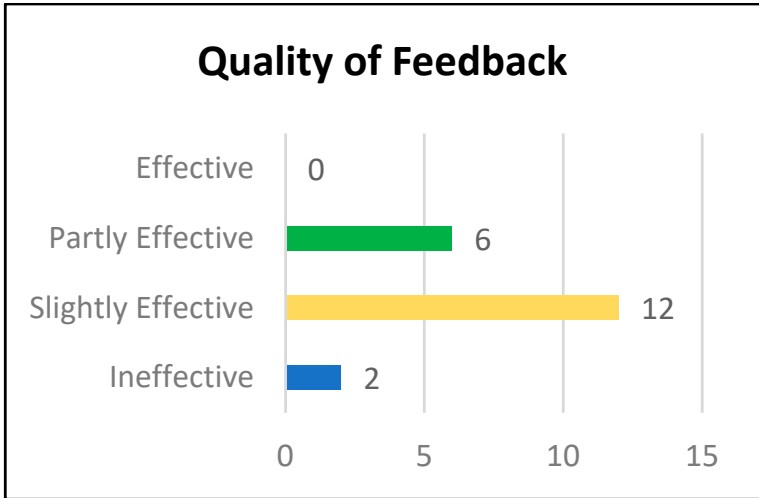

**Figure 10.** Quality of feedback in Learning Activity 2.

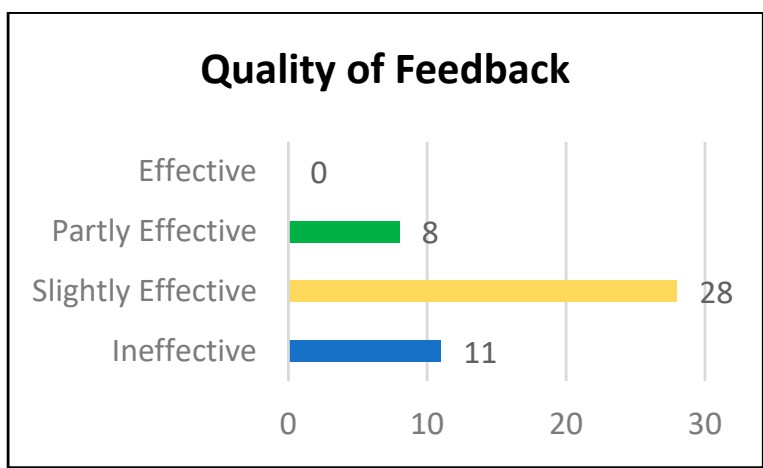

**Figure 11.** Quality of feedback in Learning Activity 3.

One essential feature of effective feedback is to offer opportunities for improving learning [14]. The rubric used in the current study to evaluate the quality of the feedback included "opportunities for improving learning" in the effective and partly effective categories. The percentage of effective and partly effective feedback responses together in each learning activity was 40% or less. The following two examples of feedback illustrate the typical nature of the feedback in these two categories:

- Good example: Procedural understanding is when students carry on steps and algorithms and here they follow steps to count units and add numbers to find perimeter and multiply them to find area, so they rarely make deep connections during instruction.
- I think both examples assess Procedural understanding, you could find a non-routine problem to assess conceptual understanding. [Effective] (Learning Activity 1).
- I agree with you that it is routine because the student used to solve this type of question during the lesson but I think it is procedural because you need to follow a procedure to find the perimeter. [Partly Effective] (Learning Activity 2).

In these examples, the students challenged their peer's thinking either by offering a suggestion that included completing the assignment using a different approach (example 1) or by not agreeing with the peer and offering an alternative perspective (example 2).

The feedback that fell into the ineffective category was typically brief affirming or praising communications:

- Very clear examples and reasons. [Ineffective] (Learning Activity 1).
- I agree with you, to specify the student who is required to understand the four concepts. [Ineffective] (Learning Activity 2).
- The question is clear and understandable for the student. [Ineffective] (Learning Activity 3).

This type of feedback did not include comments about future actions that the peer could use to deepen their understanding. The feedback in the slightly effective category was similar to the ones in the ineffective category. They differed in that slightly effective feedback included some justification or explanation about the task/concepts:

- A good question that reflects the student's understanding of the concepts of division, multiplication, and the connection between them. [Slightly Effective] (Learning Activity 3).
- Yes, the students would be able to understand. The pictures help students understand and solve the problem. [Slightly Effective] (Learning Activity 3)

## 4. Discussion

The function of feedback is that it has a key role in peer communications as it directs the cognitive level of the interactions [20]. It also determines the length and number of communications. For example, a feedback response that includes questions to a peer might trigger further responses from the peer, increasing the possibility of further reflective communications [46]. The data analysis of the current study revealed that the participants used a variety of types of communication in their peer feedback. It was important that they provided justifications or explanations along with most of their affirmation/negations or praise. Elaborating ideas in feedback responses and offering suggestions are found less threatening and more formative in nature [6,47]. Although not to a great extent, the participants elaborated their ideas and offered suggestions to their peers to enhance their work in some of the communications. Questioning, on the other hand, was observed only once. This type of communication has a reflective function in feedback and promotes deeper and lengthier analysis by the students [3]. Reflective questioning can promote the self-regulation of learning [48] and thus is an important type of communication that should take place and be promoted in peer feedback.

With respect to the level of feedback, the data analysis revealed that none of the feedback focused on the person (self) who answered the discussion prompt. Feedback concentrating on self may not lead to the achievement of learning objectives [7,14]. To be constructive, feedback should focus on tasks, processes, or self-regulation. In the current study, the participants focused on the task when providing feedback to their peers. There was little feedback in the process category and none in the self-regulation category. Feedback at the task level might show immediate effects on a student's performance while feedback at the process level might help the student develop strategies that can be used in future learning [7]. Therefore, it is positive that the participants used task and process-related comments in their feedback. However, more process and self-regulation-oriented feedback practices are needed for the students to develop skills that will enable them to take control and assess their own learning processes [6,49].

Regarding the quality of the feedback, the findings showed that less than half of the feedback responses fell into the effective and partly effective categories. High-quality feedback should communicate the students' current performance and indicate approaches that could be used to reach the desired performance, offering opportunities for monitoring and regulating learning [14,16]. Higher education students appreciate and value this type of constructive feedback [49]. In the current study, although some students showed the capacity and performance required to offer high-quality feedback, there seems to be a need to support the students in providing effective and constructive feedback. Taking into consideration that peer feedback in online learning activities is gaining more importance due to the widespread use of online courses, instructors should consider approaches designed to improve the quality of feedback developed and shared by students.

It is important to note that the participants of the current study did not receive any structured training on how to offer peer feedback. Nor were they given guidelines on how to write effective feedback. Our goal was to explore the existing status of the types of feedback developed and used by graduate students for diagnostic purposes. In this context, we were able to understand the strengths and weaknesses of the feedback given by our students through engaging in this study. Similar patterns may be observed in other online learning settings as well [1,31]. In this context, we found that our students need support with improving the quality of their feedback, such as asking reflective questions, elaborating their justifications, and providing process and self-regulation-oriented comments. The lack of feedback in these areas might partly be due to language and cultural factors. The participating students provided feedback in English, which is not their first language. Allen and Mills [50] showed that language proficiency significantly influences the quality of feedback. Another cultural aspect is related to criticizing others' work. Students might have focused on positive aspects of their peer's work due to thinking that feedback is equivalent to criticizing [34]. Future research may take language and cultural aspects into

account by allowing students choice in terms of the language used whilst providing peer feedback and clarifying that feedback is a critical process completed to help improve their peer's work and learning. A focus on using questioning feedback might be helpful in this regard.

The current study contributes to the existing literature on peer feedback in several ways. First, this study provides an analytic approach that can be used to examine the nature of peer feedback from three different perspectives: function, level, and quality. Previous research typically focused on one or two aspects of feedback nature [e.g., 6,7,14,18,19,38]. Future research may benefit from the codes used in this study to examine feedback offered by students in this context. As the number of online/blended courses are increasing recently, the availability of such an analytic tool is timely and valuable. Researchers from other regions or countries may use or adapt the data analysis tools developed in the current study. Second, the current study contributed to the limited number of research studies on peer feedback developed and used by graduate students. Finally, there is scarce research on peer feedback in the UAE. This is the first study that has shed light on the function, level, and quality of peer feedback in the UAE context. In fact, one of the key facets of our research lies in its focused attention on the cultural context of our participants. Our investigation delves into the process of giving and receiving peer feedback, a practice that has not traditionally been prevalent within the cultural framework of the participants. By examining the nature of peer feedback within this unique cultural setting, our study aims to contribute to the current understanding of novel approaches to peer feedback within diverse cultural contexts. The distinctiveness of our participant demographics, comprising Emirati individuals, offers a significant contribution to the existing body of literature on peer feedback. This contribution manifests through the valuable insights it provides into the type and quality of peer feedback specifically within the precise cultural and regional context of the Emirati people.

## 5. Conclusions

As future computers and technology become more intelligent and learning opportunities are provided through advanced interactive software, offering blended courses or fully online courses will be more prevalent in all facets of education, and assessment tasks might become more complex. The need to encourage higher education students to be more skilled in providing meaningful and high-quality peer feedback will increase. In the current study, the function, level, and quality of peer feedback have been investigated. Based on the findings of the current and prior studies [20,34], there seems to be a need to support students in providing effective and constructive feedback. Taking into consideration that peer feedback in online learning activities is gaining more importance due to the widespread implementation of online courses, higher education institutions should include modules of practices and approaches to improve the quality of feedback developed by its students. The findings of the current research can be used to identify areas of improvement that can be targeted in these modules or interventions. Future research might investigate the design and implementation of such modules to find out the best practices that could be used to improve the quality of peer feedback.

**Author Contributions:** Conceptualization, E.E., J.M.R., A.M.J. and J.D.J.; methodology, E.E. and J.M.R.; writing—original draft preparation, E.E., J.M.R., A.M.J., J.D.J. and S.M.G.; writing—review and editing, E.E., J.M.R., A.M.J., J.D.J. and S.M.G. All authors have read and agreed to the published version of the manuscript.

**Funding:** This research received no external funding.

**Institutional Review Board Statement:** The study was conducted in accordance with the Declaration of Helsinki, and approved by the Institutional Review Board of Emirates College for Advanced Education (RP-165-2022, 26-April-2022).

**Informed Consent Statement:** Informed consent was obtained from all subjects involved in the study.

**Data Availability Statement:** The data presented in this study are available on request from the corresponding author with restrictions due to Institutional Review Board rules.

**Conflicts of Interest:** The authors declare no conflict of interest.

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
