# Peer review of "Exploring the Type and Quality of Peer Feedback in a Graduate-Level Blended Course"

_education, doi:10.3390/educsci13060548_

Round 1

Reviewer 1 Report

Authors made a great and original work. Tha analysis is conducted well and it is useful to evaluate the feedback about a blended course.

I suggest the acceptance for consideration of this paper.

Author Response

Dear Reviewer,

We appreciate your opinion on our manuscript and are grateful for your feedback. Thank you very much. 

Sincerely,

Authors

Reviewer 2 Report

As the authors state, this study has the potential to add to the understanding of local education and learning in higher education.

I have comments about four areas of the report. First, there is a lack of attention given to issues and implications of sampling. The participants volunteered. Implications?  Do we know more about the sample? Males/Females? Age range? Ability?

Second, were the discussion prompts trialled for refinement of clarity and reliability?

Third, there are some matters of style. In some sections of the paper (e.g. lines 108-110, 151-156, 167-169) there is a lapse from objectivity.

Finally, there has been no differentiation in the use of sources based on school children rather than higher education students. This needs to be justified or the lack of distinction acknowledged.

In summary, though, this paper merits publication. It has a contribution to make and the authors have clearly shown this in their report of their findings.

Author Response

Dear Reviewer 2,

We appreciate your opinion on our manuscript and are grateful for your feedback. We believe that the feedback we received enabled us to improve the article, and hope you concur and find it worthy of publication in the Education Sciences. In the attached file, we explain how we addressed your suggestions. We will be happy to address further concerns.

Kind regards,

Authors

Reviewer 3 Report

Although certain teaching practices and data analysis have been carried out, due to the lack of research innovation, and no exciting new conclusions or methods have been seen, it belongs to the type of repetitive research. Therefore, the following suggestions are proposed:

1. It is recommended to modify the topic and add educational innovation features

2. It is recommended to focus on the characteristics of local higher education, highlighting educational differences caused by regional or cultural backgrounds, rather than repeating similar topics

3. I am not very clear about the research value and significance of this research conclusion for other regions or countries, so it is recommended to highlight this part of the content

4. Research methods need to be improved, such as adding credibility analysis and visual analysis

Author Response

Dear Reviewer 3,

We appreciate your opinion on our manuscript and are grateful for your feedback. We believe that the feedback we received enabled us to improve the article, and hope you concur and find it worthy of publication in the Education Sciences. In the attached file, we explain how we addressed your suggestions. We will be happy to address further concerns.

Kind regards,

Authors

Round 2

Reviewer 3 Report

The author has partially revised the research content, which effectively solves the problems in research analysis and result description. However, due to the limitations of research goal setting, it is also difficult to have further innovation. Looking forward to surprising new discoveries in teaching tools, environments, or methods in the future.